# Chios Mastic Gum: A Promising Phytotherapeutic for Cardiometabolic Health

**DOI:** 10.3390/nu16172941

**Published:** 2024-09-02

**Authors:** Sarah A. Blomquist, Maria Luz Fernandez

**Affiliations:** 1School of Nutritional Sciences and Wellness, The University of Arizona, Tucson, AZ 85721, USA; sblomquist@arizona.edu; 2Department of Nutritional Sciences, University of Connecticut, Storrs, CT 06269, USA

**Keywords:** Chios mastic gum, mastiha, cardiometabolic protection, antioxidant, anti-inflammatory

## Abstract

Chios mastic gum (CMG) is a resin obtained from the *Pistacia lentiscus* var. *Chia* tree that grows in the Mediterranean. For millennia, it has been renowned for its medicinal properties, but recently, CMG has gained attention due to its pronounced anti-inflammatory and antioxidative properties and its use in oral health, inflammatory bowel disease, cancer, and risk factors related to cardiovascular and metabolic diseases. This narrative review seeks to briefly overview its bioactive constituents and examine and describe its potential as a cardiometabolic disease (CMD) phytotherapeutic. The results of clinical trials and in vivo, in vitro, and in silico studies provide accumulating evidence of the mechanisms underlying CMG’s impacts on lipid and glucose metabolism, cardiovascular and hepatic health, inflammation, oxidative stress, body composition, and microbiota. Despite the relatively limited studies with mixed results, they have provided the foundation to understand the strengths, weaknesses, and opportunities moving forward that may help to establish CMG and its bioactives as viable therapeutics for CMD.

## 1. Introduction

### History and Traditional Uses of Chios Mastic Gum

The mastic tree (*Pistacia lentiscus* L. var. *latifolius Coss* or *Pistacia lentiscus* var. *Chia*) grows in the region of Chios, a Greek island nestled in the northern Aegean Sea. This island is renowned for its production of a medicinal and aromatic resin known as Chios mastic gum (CMG). CMG is derived only from the trunk and branches of this tree, despite the fact that *Pistacia* species are distributed throughout the Mediterranean basin. CMG cultivation by 24 “mastic villages” is a centuries-old cultural heritage practice and tradition. From July to August, incisions are made in the bark with a tool which allows the resin to weep out and solidify into “tears”, which are then manually collected. In addition to CMG’s use in culinary, religious, and cultural practices, it has also been valued for centuries for its therapeutic applications. CMG is known as a traditional medicine, where the prominent Greek physician Hippocrates (4th century BC) and later Dioscorides and Galen (1st and 2nd century AD) noted its therapeutic properties, such as for indigestion and blood and respiratory problems [1]. Mastic gum’s use in “curing” peptic ulcers and providing protection against *Helicobacter pylori* rapidly entered mainstream academic literature after a 1998 *New England Journal of Medicine* publication [2]. Since then, research has continued to reveal the rich collection of antimicrobial, anti-inflammatory, and antioxidant properties within CMG, rendering it a versatile therapeutic agent for addressing various health conditions.

Chios mastic gum consists of the polymer cis-1,4-poly-β-myrcene (approx. 25%) [3], a volatile fraction (also known as essential oil [EO], approx. 2%), and a triterpenic fraction (approx. 65–75%) [4]. The EO’s composition is hydrocarbons (50%), oxygenated monoterpenes (20%), and sesquiterpenes (25%) [4]. Numerous groups have extensively studied the chemical composition of the EO [5,6,7,8,9,10,11], primarily by gas chromatography mass spectroscopy (GC-MS), and approximately 70 constituents have been identified [11,12]. Notably, 65–85% of the EO’s weight consists of the compounds α-pinene (30–75%) and β-myrcene (3–60%), with minimal levels of β-pinene (1–3%), linalool, trans-caryophyllene, and camphene, and a nominal contribution from an array of other compounds [13]. The triterpenic fraction consists of neutral (27%) and acidic groups (38%) and establishes the main bulk of CMG [14]. Isomasticdienonic, mastadienoninc, oleanolic, and moronic acids have been the focus of most chemical characterization studies [4,11,15], though the investigation for new triterpenoids continues [11,16]. CMG also contains traces of phenolic compounds such as vanillic, gallic, *trans*-cinnamic, *o*-coumaric, and protocatechuic acids [17]. A more complete list of the primary constituents in CMG (not an exhaustive list) can be found in Table 1. 

The utilization of tree resins and their constituents for therapeutic purposes has been recognized for millennia [18,19,20,21,22,23,24,25,26]. Among the numerous applications of resin-derived medicines, research has primarily attributed their antioxidant and anti-inflammatory properties to the mitigation of disease [25,27,28]. CMG and its array of bioactive molecules are chiefly recognized for their antioxidant and anti-inflammatory activities as well, which have been reviewed extensively [13,14,29,30,31,32,33]. Previous literature reviews have also addressed CMG’s application specifically in gum and oral health [34,35], inflammatory bowel disease [36], and CMD risk factors [37], and for its anticancer potential [38].

In addition to the antioxidant and anti-inflammatory potential that has been more broadly demonstrated by CMG, emerging evidence demonstrates that CGM impacts the clinical markers related to cardiovascular and metabolic diseases. Cardiovascular disease (CVD) is considered the leading cause of death worldwide [39], while the prevalence of metabolic diseases has generally been increasing over the past two decades [40]. There are several metabolic alterations that predispose an individual to CMD, including dyslipidemias [elevated plasma LDL-cholesterol (LDL-C) and triglycerides (TGs) and low concentrations of HDL-cholesterol (HDL-C)] [41], small dense LDL characterized for its high susceptibility to oxidation [42], high blood pressure [43,44], inflammatory conditions [45,46,47], and oxidative stress [48,49]. Clinical studies where the efficacy of CGM for cardiometabolic disease has been demonstrated, as well as the mechanisms involved in its CMD protective effects from the evidence derived from animal studies, cell cultures, and pharmacophore models, are discussed below.

To conduct this narrative review, a bibliographic search was conducted in Google Scholar, PubMed, and ScienceDirect (up to May 2024) using combinations of the following keywords: Chios mastic gum, mastiha, *Pistacia lentiscus* var. *Chia*, *Pistacia entiscus* L. var. *latifolius Coss*, cardiometabolic. Only articles written in English with the full text available and those that addressed aspects of cardiometabolic disease were considered, resulting in the inclusion of 20 articles for the narrative literature review.

## 2. Evidence for the Utility of Chios Mastic Gum in the Treatment and Management of Cardiometabolic Disease

### 2.1. Human Studies

The effects of CMG supplementation on lipids and lipoproteins, hepatic, cardiovascular, and general metabolic health has been explored in clinical trials and is described below. One of the earliest reports [50] evaluated the hypolipidemic effects of CGM when given at a high dose (5 g/d) (*n* = 48) and observed decreases in plasma cholesterol (TC), TG, TC/HDL ratio, lipoprotein (a), apolipoprotein A-1, and apolipoprotein B after 18 months. In contrast, a group of patients who received the low dose (~0.7 g/d) (*n* = 85) for 12 months exhibited no changes in lipids. As a result of high-dose CMG supplementation, decreases in serum alanine (ALT), aspartate transaminase (AST), and gamma-glutamyl transferase were also observed, indicating the hepatoprotective properties of CMG. Differential sex effects were discovered as well, where beneficial effects of CMG on total cholesterol (TC), lipoprotein (a), and serum glucose (in the low-dose group) occurred in males. This study suggests high doses, longer duration, and sex play a role in the outcomes of CMG supplementation.

Fukawaza et al. [51] investigated the effects of mastic powder (MP) (5 g/d) compared to a placebo or to mastic powder plus physical activity (measured as steps/day) (MP + PA) in a limited population of 21 Japanese men (aged > 40) at 3 and 6 months. MP and MP + PA reduced serum TG as early as 3 months when compared to controls. In addition, MP alone reduced insulin and HOMA-IR values at 6 months, while MP + PA potentiated the effects and reduced insulin and HOMA-IR as early as 3 months. However, no changes were observed in body composition, blood pressure, liver function, or other lipid parameters.

Kartalis et al. [52] analyzed data from 156 hypercholesterolemic subjects (TC > 200 mg/dL) to evaluate the effects of CMG or its individual components on plasma lipids and fasting plasma glucose (FPG) levels. They allocated subjects to four groups: control, CMG, polymer-free CMG, or powdered CMG. Interestingly, reductions in both TC and FPG were only observed in those individuals who consumed the whole CMG, indicating that the beneficial effect of CMG is potentially strengthened by the combination of polymeric, essential oil, and triterpenic fractions. No changes were observed in plasma LDL-C, HDL-C, and TG, or secondary measures such as uric acid or C-reactive protein (CRP) [52].

A recent study by Gioxari et al. [53] recruited 94 subjects who were metabolically unhealthy and were allocated to either the daily consumption of 200 mg CMG EO or a control group for 3 months. Those subjects allocated to the GMC group exhibited a lowering of plasma TG and LDL-C when compared to the control group. Additionally, improvements in anthropometric parameters, such as weight, percentage of body fat and visceral fat, and body mass index, and in hepatic and cardiovascular markers, such as ALT and systolic blood pressure (SBP), were demonstrated in the treatment group. The investigators also observed beneficial effects on oxidation as documented by decreases in oxidized LDL, a major biomarker of oxidative stress and atherosclerosis [54], and increases in adiponectin, an adipokine that protects against inflammation and type 2 diabetes [55]. No changes were observed in inflammatory markers such as tumor necrosis factor alpha (TNF-α), CRP, or interleukin-6 (IL-6), although there were improvements in the self-reported quality-of-life measures [53].

The MAST4HEALTH study that was conducted at three clinical trial sites (Greece, Italy, and Serbia) evaluated the effects of CMG in people diagnosed with non-alcoholic fatty liver disease (NAFLD) [56,57]. For these studies, obese patients with NAFLD were randomly recruited and allocated to CMG (*n* = 41) or a placebo (*n* = 57) for 6 months. Notably, the MAST4HEALTH study employed nutritional counseling to allow for body weight regulation up to 5% of initial body weight. Amerikanou et al. [56] evaluated NAFLD severity utilizing magnetic resonance imaging, conducted metabolomics and a microbiota analysis, and measured clinical biomarkers to examine the associations with CMG treatment [56]. For the NAFLD severity results, only individuals with a BMI > 35 kg/m^2^ demonstrated an improved iron-corrected T1 and liver inflammation fibrosis score compared to the control. CMG also improved microbiota dysbiosis by decreasing the number of *Flavonifractor*, a microorganism known to cause inflammation. However, no differences in clinical lipid or metabolic markers were observed when the treatment and control groups were compared. The metabolomic analysis revealed significant reductions in lysophosphatidylcholine (LPC), lysophosphatidylethanolamine (LPE), and cholic acid in the CMG group. Notably, LPC stimulates the release of hepatic extracellular vesicles from hepatocytes which contain pro-inflammatory cargo and ultimately stimulate inflammatory cascades [58]. LPC also serves as a marker of PC depletion and is associated with membrane dysfunction, lipotoxicity and inflammation [59,60]. Altogether, the data suggest the improvements from CMG in individuals with severe obesity may be due to interactions between gut microbiota, bile acid synthesis, and lipids and their impact on energy metabolism [56].

Kanoni et al. [57] also examined the MAST4HEALTH cohort and reported a higher total antioxidant status compared to the placebo but only in those patients that had a BMI >35 kg/m^2^. No other differences between biomarkers as a result of CMG supplementation alone was shown. However, numerous CMG–gene interaction associations with cytokines and antioxidant biomarkers were discovered. Some of the genetic loci are involved with NAFLD pathways, such as the lanosterol synthase (*LSS*), the mitochondrial pyruvate carrier-1 (*MPC1*), the sphingolipid transporter-1 (*SPNS1*), the transforming growth factor-beta-induced gene (*TGFBI)*, the micro-RNA 129-1 (*MIR129-1*), and the granzyme B (*GZBM*) genes [57]. This study underscores the potential downside of comparing CMG to placebo groups based on clinical biomarkers alone and the potential impact of nutrigenetic interactions in utilizing botanical supplements for cardiometabolic disease.

Kontogiannis et al. [61] also investigated the effect of CMG on blood pressure in a short-term study. Twenty-seven participants (including thirteen hypertensive) were recruited for a cross-over study and were randomly allocated to 2.8 g of CMG or a placebo to be taken one week apart. Two to three hours after administration, blood pressure and the aortic augmentation index were measured, as well as the expression of several genes related to pro-oxidant responses and pathways of inflammation. Improvements in peripheral and aortic SBP and peripheral pulse pressure were observed in hypertensive subjects after CMG administration, and the gene analysis indicated a downregulation of the *NOX-2* pro-oxidant pathway [61]. No improvements were observed in the normotensive subjects, suggesting CMG impacts hemodynamic parameters through effects on genes involved in proteostatic and pro-oxidant pathways.

Table 2 summarizes these findings, where CMG supplementation studies (from 8 weeks to 18 months) in individuals ranging from healthy to metabolically unhealthy demonstrate modest improvements in cardiometabolic risk markers, cytokines, antioxidant and inflammatory status, and microbiota diversity. The greatest improvements were seen in individuals with more severe obesity, those who were hypertensive, had longer supplementation regimes, or were allocated to higher doses (Table 2). Notably, none of these clinical trials reported any side effects with CMG supplementation, making it a safe and potentially desirable therapeutic.

### 2.2. Animal Studies

The effects of CMG alteration on lipids, liver, heart, and general metabolic health has been more rigorously demonstrated in disease models in animals and is described below. In a 2011 study by Vallianou [62], hyperlipidemic-induced and naïve rats were treated with four doses (2.5%, 4%, 5%, and 7.5%) of mastic gum essential oil (CMG EO) or a primary constituent of CMG EO (~ 0.83%), camphene, at 1.5–3.0 µg/g body weight. CMG EO treatment in naïve rats resulted in cholesterol and TG reductions, where a 4% CMG EO treatment reduced them by 54% and 30%, respectively. Hyperlipidemic rats exhibited a stronger hypolipidemic effect, where in the 4% MGO treatment group, cholesterol was reduced by 56%, LDL-C by 52%, and TG by 50%. In both naïve and hyperlipidemic rats, dose-dependent effects were observed. Treatment with 30 µg/g camphene in hyperlipidemic rats also resulted in significant reductions in cholesterol, LDL-C, and TG by 54%, 54%, and 34%, respectively [62].

Georgiadis et al. [63] observed that after 8 weeks of CMG supplementation in streptozotocin-induced diabetes mice, the low-dose group (20 mg/kg body weight) presented lower cholesterol, LDL-C, and TG and improved HDL [63] compared to the baseline, while the high-dose group (500 mg/kg body weight) revealed reduced plasma TGs only compared to the control [63]. One significant confounder noted by this study was the exact daily food and CMG consumption by each animal was not possible to determine.

In 2016, Andreadou et al. [64] subjected rabbits to ischemia and provided 6-weeks supplementation of mastic extract without polymer (ME) and the mastic neutral fraction (MN), followed by reperfusion. Both ME and MN administration to hypercholesterolemic rabbits resulted in hypolipidemic effects via reductions in cholesterol and LDL-C by 47% and 88%, respectively. A 2019 study by Kannt et al. [65] also induced non-alcoholic steatohepatitis (NASH) in mice and supplemented CMG (0.2% weight for weight) for 8 weeks. They reported that CMG supplementation resulted in lower liver cholesterol and total lipids compared to untreated controls.

The effects of CMG on liver health are also documented in a few studies where, for example, hepatic steatosis is partially reversed in both low- and high-dose CMG groups compared to the control [63]. Kannt et al. [65] also demonstrated as a result of CMG supplementation reduced alanine transaminase (ALT), improved hepatic steatosis via reduced fibrosis (Col1a1 and galectin-3 as biomarkers), and a reduced NAFLD score.

CMG on heart health was investigated by Tzani et al. [66] in 2018 using 2-kidney, 1-clip (2K1C) hypertensive rats that were treated with CMG at a dose of 40 mg/kg body weight/day for 2 weeks. When comparing 2K1C to 2K1C + CMG rats at the end of treatment, reductions in SBP, DBP, mean arterial pressure, and renin were observed. 2K1C + CMG rats also showed mitigation of target organ damage as illustrated by improvements in biomechanical properties of the aorta, such as reductions in slope, heart weight, cardiac tissue small vessel hypertrophy, wall thickness, and cross-sectional area, and median area to lumen area ratio. Andreadou et al. also demonstrated ME and MN significantly reduced infarct size, however, only in the normal-fed and not the hypercholesterolemic rabbits [64].

The impacts of CMG on general metabolic markers were also briefly examined in these studies, where Georgiadis and colleagues [63] observed decreased serum glucose levels after 4 weeks in both low- and high-dose CMG compared to the baseline. In Tzani et al., the anti-inflammatory actions of CMG were demonstrated by decreases in CRP and IL-6 compared to the hypertensive animals that did not receive CMG treatment and were sustained for up to 10 weeks after the follow-up [66]. Lastly, Kannt et al. measured changes in microbiota diversity with CMG supplementation and suggested it promoted a partial but significant recovery of diversity (*p* = 0.0496) [65]. Diversity measures included Faith’s phylogenetic diversity, number of OTUs, and Shannon’s diversity index. Of note, the microbiota composition correlated significantly with the steatosis score, NAFLD score, % liver lipids, liver TG, plasma ALT, and genetics [65].

Table 3 summarizes these findings, where CMG supplement studies ranging from 24 h to 8 weeks in animal models exhibit improvements in biomarkers predominantly related to lipid metabolism, as well as in hepatic health and disease, heart biomechanical indices and cardiovascular parameters, and gut microbiota diversity. Disease severity influenced the efficacy of CMG treatment (where increased severity could result in improved or poorer outcomes, depending), as did the duration and dosing regimen.

### 2.3. Proposed Mechanisms of Action from In Vitro and In Silico Studies

Several pathways and mechanisms have been purported to explain CMG’s benefit specifically on cardiometabolic outcomes using cell cultures (in vitro) and pharmacophore (in silico) studies. These findings can be summed up as CMG modifying inflammation and immune responses, oxidative stress and antioxidant responses, and metabolic regulation and signaling. Loizou et al. [67] examined the effect of a CMG neutral extract and tirucallol (a triterpene) on the adhesion molecule expression and attachment of monocytes in TNF-α-stimulated human aortic endothelial cells. Both treatments inhibited vascular cell adhesion molecule 1 (VCAM-1) and intercellular adhesion molecule-1 (ICAM-1) expression, as well as the binding of U937 cells, and also attenuated the phosphorylation of NF-κB p65. Their data suggest CMG may mitigate atherogenesis through alterations to endothelial cells and NF-κB-mediated inflammatory cascades.

In Vallianou’s 2011 [62] publication, HepG2 cells were also treated with camphene or mevinolin (a statin) in order to examine effects on cholesterol biosynthesis. After 1 h of incubation, cholesterol content was decreased by 20% in both conditions compared to the control, while an 18 h incubation demonstrated a greater reduction by mevinolin. HMG-CoA enzyme activity assays also revealed cholesterol reduction via camphene occurred independently of HMG-CoA reductase activity [62].

Human keratinocyte cells were utilized by Xanthis et al. [68] to evaluate the effects of CMG EO and its primary constituents on antioxidant and cytoprotective potential. While the EO did not demonstrate direct free radical scavenging, it did alter gene expression profiles of oxidative-response-related genes *NRF2*, *GSTP1*, *SOD1*, *NQO1*, *GPX1*, *HMOX1*, and *CAT* [68]. CMG EO and its primary constituents also enhanced cell viability after exposure to H_2_O_2_ and UVB, as well as enhanced cell migration and wound closure in a scratch assay, highlighting the cytoprotective and regenerative potential of CMG.

In a pharmacophore-based virtual screening study by Peterson et al. [69], oleanolic acid and other subfractions of CMG were identified as partial agonists for PPARγ. Transient transfection experiments with oleanolic acid demonstrated PPARγ agonism reaching 20% that of rosiglitazone (a full agonist) [69]. Notably, partial PPARγ agonists are currently under research and development for their utilization in type 2 diabetes mellitus [70]. In another pharmacophore study, Vuorinen and colleagues [71] sought to identify 11β-hydroxysteroid dehydrogenase 1 (11β-HSD1) inhibitors, which have previously been proposed to combat metabolic disorders. The main triterpenoids in CMG’s acidic fraction, masticadienonic acid and isomasticadienonic acid, were identified to inhibit 11β-HSD1 (up to 50% of enzyme activity) at low micromolar concentrations (2 µM), suggesting these compounds might be contributing to the antidiabetic actions of CMG [71]. 

A 2004 study by Dedoussis et al. [72] demonstrated that a CMG polar extract mitigated the typical response of apoptosis and necrosis of peripheral blood mononuclear cells (PBMCs) when they were cultured with oxidized LDL (oxLDL). They observed CMG restored glutathione (GSH) levels and downregulated CD36 levels. A deeper investigation led the authors to conclude it was primarily CMG’s triterpenic fraction acting to upregulate the antioxidant defenses of PMBCs (rather than acting directly on oxLDL), such as through GSH and CD36 pathways [72]. A 2003 study by Andrikopoulos et al. [73] also investigated the protective effects of CMG and its constituents on the copper-induced oxidation of human LDL, where CMG EO and its fractions exhibited a protective effect on LDL ranging from 65.0% to 77.8% [73]. 

In a recent study, Kalousi et al. [74] examined the impacts of three subfractions (apolar, medium-polar, polar) of CMG in HEK293 cells on the cell viability and transcriptional activity of molecules involved in cellular energy homeostasis. In contrast to a previous study demonstrating CMG’s enhancement of cell viability after oxidative stressors [68], this study demonstrated CMG subfractions alone reduced cell viability in a dose-dependent manner, promoting mitochondrial-induced apoptosis and antiproliferative activities through reductions in procaspase and blc2-proteins. Medium-polar (primarily triterpenes) and polar fractions also dose dependently suppressed dexamethasone-induced glucocorticoid (GR) transcriptional activation, GR protein levels, and adenosine monophosphate-activated protein kinase alpha (AMPKα) phosphorylation, suggesting CMG’s role in numerous physiologic processes such as immunity and metabolism. Additionally, medium-polar and apolar fractions also reduced the protein levels of GR target genes including phosphoenolpyruvate carboxykinase (PEPCK) and PPARα, which are key genes involved in glucose and lipid metabolism, respectively. The medium-polar fraction also exhibited the highest anti-inflammatory actions via the dose-dependent suppression of TNF-α-induced NF-κΒ transcriptional activation and a reduction in the p65 NF-κΒ subunit.

Table 4 summarizes these findings, where initial investigations have demonstrated across a variety of cell culture models that CMG’s therapeutic potential may be driven by its impact on inflammation and immunity (adhesion molecules, monocytes, NF-κB, cell migration), transcription factors and signaling pathways (NF-κB, NRF-2, GR, PPARγ/α, AMPKα), oxidative stress and the antioxidant responses (NRF-2, GSH), lipid metabolism and cardiovascular health (oxLDL, CD36, PEPCK), and cell survival (cytoprotection, apoptosis).

## 3. Critique and Future Directions

The impact of CMG on cardiometabolic markers is not consistent among the limited number of human studies. While some show hypolipidemic effects and improvements in metabolic, antioxidant, and inflammatory markers, others show no differences compared to a placebo. This is likely due to differences in the type of population studied (healthy versus metabolically unhealthy), the dose and format administered (EO vs. CMG vs. CMG constituents), the experimental design (duration, dietary, or physical activity confounders), and baseline genetics and microbiomes. Shorter study durations and smaller sample sizes are among the challenges in interpreting these studies. However, some of the studies provide emerging insight into the actions of CGM, such as decreases in LPC, improved microbiota diversity, and CMG–gene interactions in subjects with NAFLD and obesity, as well as the downregulation of oxidative pathways in hypertensive subjects. More studies are needed to elucidate the conditions under which CGM is most effective to mitigate cardiometabolic disease. Specific genetic variants related to inflammation, proteostatic and pro-oxidation pathways, as well as sex, appear to influence differential responses to CMG and should be considered in future clinical trials [57]. Increased disease severity also appears to enhance CMG’s therapeutic impacts, which should be considered in the recruitment of subjects [52,56,57]. Given the relationship between diet, energy metabolism and homeostasis, and cardiometabolic disease, [75,76,77], dietary intake and physical activity should also be accounted for in future clinical trials, as they are significant confounders. A possible method to potentiate the positive impacts of CMG, however, may include utilizing it alongside Mediterranean-style, plant-based, or Nordic dietary patterns.

Other lifestyle factors (such as smoking, alcohol consumption, sleep patterns), the dosing regimen of CMG, interactions with potential medications, and underlying health conditions are additional confounders that have yet to be fully acknowledged and addressed in these clinical trials. These mixed results and study designs underscore the need for more rigorous research to validate the therapeutic efficacy of CMG.

In vivo, in vitro, and in silico studies provide promising insight into CMG’s therapeutic actions. Animal models targeting a specific disease pathology (diabetic, hypertensive, hyperlipidemic/cholesterolemic, and NASH) demonstrated improvements in biomarkers and mechanisms that are translatable to human studies, such as lipids, glucose, blood pressure, renin, inflammatory markers (CRP, IL-6), and hepatic and cardiovascular markers. Similar to the human studies, higher administered doses of EO enhanced CMG’s effects in Vallianou et al., who also observed there is a “cut off” dose for the therapeutic effects with hyperlipidemia, where an increased dose no longer provides a substantial benefit [62]. In fact, Georgiadis et al. revealed the limited benefit of CMG in the high-dose group compared to the low-dose group in a diabetes model; however, their study noted a significant confounder in terms of accounting for CMG intake [63]. In contrast to the human studies where increased CMG enhanced beneficial effects, these animal studies raise the possibility that certain doses or preparations of CMG may be used to target specific disease states or mechanisms, and that a “one size fits all” approach may not be appropriate. The results from Andreadou et al. [64] also highlighted CMG’s enhanced benefits (on lipid peroxidation and infarct size) for normal-fed rabbits and not hypercholesterolemic, suggesting differential therapeutic mechanisms depending on disease severity [64].

Cell culture and pharmacophore studies provide the groundwork to suggest the mitigation of CMG on cardiometabolic disease may be occurring primarily through its effects on inflammation/immunity, oxidation/antioxidants, and metabolic regulation and signaling. While many botanical medicines work on inflammatory and antioxidative mechanisms, findings that CMG potentially promotes beneficial effects through GR and PPARγ/α are of particular interest, especially given the current development of pharmaceuticals that work through those mechanisms to mitigate cardiometabolic disease [69,71,74,78,79,80]. Studies examining the medicinal potency of CMG subfractions and CMG’s constituents (polymer, EO, triterpenes) are also valuable. While the triterpenic oleanolic, masticadienonic, and isomasticadienonic acids and EO fractions have generally received more attention, there is evidence to suggest a unique synergy between EO compounds as well as CMG’s constituents [69,73]. CMG’s potential interactions with other phytochemicals and pharmaceuticals may be another fruitful route of investigation.

Figure 1 summarizes the observed mechanisms of CMG’s impact on cardiometabolic outcomes and disease mechanisms in human, animal, in vitro, and in silico studies. For metabolic and microbiota modulation, human studies have revealed CMG reduces TG, LDL, TC, insulin, fasting glucose, HOMA-IR, Lp(a), Apo(B), LPC, LPE, and cholic acid. Increases in HDL and microbiota diversity were also observed. In adjunct metabolic syndrome treatments, there was a reduction in body fat, visceral fat, and weight and an increase in adiponectin. CMG–gene interactions also revealed reductions in hemoglobin. Animal studies validate some of these findings, with reductions in glucose, TG, LDL, TC, and total lipids and increases in HDL and microbiota diversity. In vitro and in silico studies reveal these results may be partially mediated through the reduction in activity of various pathways and receptors such as PEPCK, GR, PPARγ/α, AMPKα, and 11β-HSD1.

For cardiovascular and hepatic impacts in humans, studies have shown CMG supplementation reduces SBP and pPP, while adjunct metabolic syndrome treatment studies have also shown additional reductions in ALT, AST, and GGT. Animal studies fortify these findings with the observed reductions in hepatic steatosis, NAFLD score, liver fibrosis, ALT, SBP, DBP, MAP, renin, and infarct size. Improvements in numerous cardiac indices have also been observed with CMG supplementation. Modulation in oxidative stress and antioxidant genes and pathways has been observed in humans because of CMG supplementation, such as reductions in oxLDL and NOS-2. CMG–gene interaction studies have revealed a differential impact on glutathione peroxidase (Gpx) levels (increase or decrease) and an increase in total antioxidant status (TAS). In vitro studies support these studies with reductions in oxLDL, NRF-2, CD36, and GSH.

Finally, human studies have also shown that CMG supplementation impacts inflammation and immunity, particularly through the CMG–gene axis, where a reduction in IL-6 and TNF-α and their genes was observed, as well as a reduction in IL-10. In animal studies, decreases in CRP and IL-6 have been observed, while in vitro studies have demonstrated decreases in monocyte attachment, adhesion molecules, NF-κΒ, p65, and cell migration. Notably, the modulation of oxidative stress, antioxidants, inflammation, and immunity presumably is enough to drive the cardiovascular, hepatic, and metabolic outcomes observed in human studies.

## 4. Conclusions

Accumulating evidence across human, animal, cell culture, and pharmacophore studies suggests CMG may be a unique phytotherapeutic for improving the biomarkers related to lipid and glucose metabolism, cardiovascular and hepatic health, inflammation, oxidative stress, body composition, and microbiota. Preceding research has established the groundwork and provided the clarity to further examine CMG’s impact on cardiometabolic health in clinical trials. One of the largest limitations of the human studies is the failure to fully isolate and examine the effects of CMG supplementation over the long term. Even healthy volunteers saw notable improvements in lipid biomarkers after 18 months of CMG, while shorter-term studies in metabolically unhealthy individuals showed mixed results, suggesting a longer duration may be key. Dietary intake, physical activity, and adjunct metabolic disease treatments should also be accounted for. Additional factors necessary to consider include sex, disease severity, and the influences of genetics and microbiota. Elucidating the role of the microbiota may ultimately provide valuable insights into the impacts of the gut–heart axis. Whole botanical medicines such as CMG often demonstrate greater efficacy than isolated bioactives due to the synergistic interactions of compounds; however, the exploration of individual compounds, such as EO compounds or triterpenes, also offers the opportunity for targeted therapeutic effects and novel drug development moving forward.

## Figures and Tables

**Figure 1 nutrients-16-02941-f001:**
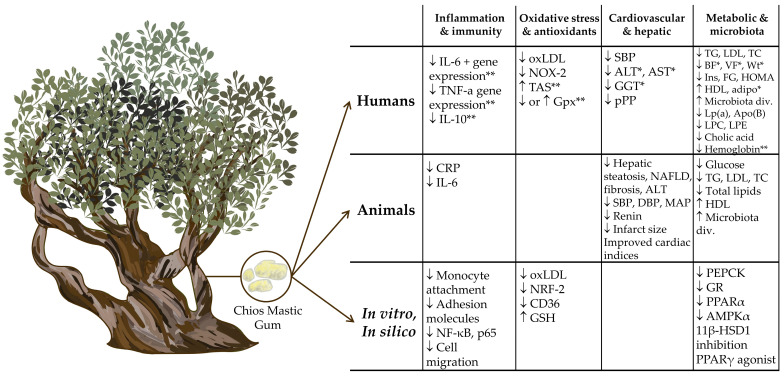
Impacts of Chios mastic gum on disease mechanisms and cardiometabolic outcomes. Abbreviations used: * adjunct metabolic syndrome treatments only, ** CMG-gene interactions only, 11β-HSD1 = 11-beta-hydroxysteroid dehydrogenase, adipo = adiponectin, ALT = alanine aminotransferase, AMPKα = AMP-activated protein kinase alpha, Apo(B) = apolipoprotein B, AST = aspartate aminotransferase, BF = body fat, CD36 = cluster of differentiation 36, CRP = C-reactive protein, FG = fasting glucose, GGT = gamma-glutamyl transferase, Gpx = glutathione peroxidase, GR = glucocorticoid receptor, GSH = glutathione, HDL = high density lipoprotein, HOMA = HOMA-IR (homeostatic model assessment for insulin resistance), IL-10 = interleukin 10, IL-6 = interleukin 6, Ins = insulin, LDL = low density lipoprotein, LPC = lysophosphatidylcholine, LPE = lysophosphatidylethanolamine, Lp(a) = Llipoprotein(a), MAP = mean arterial pressure, Microbiota div. = microbiota diversity, NAFLD = non-alcoholic fatty liver disease, NF-κB = nuclear factor kappa B, NOX-2 = NADPH oxidase 2, NRF-2 = nuclear factor erythroid 2-related factor 2, oxLDL = oxidized low-density lipoprotein, p65 = p65 subunit of NF-κB, PEPCK = phosphoenolpyruvate carboxykinase, PPARα = peroxisome proliferator-activated receptor alpha, PPARγ = peroxisome proliferator-activated receptor gamma, pPP = peripheral pulse pressure, SBP = systolic blood pressure, TAS = total antioxidant status, TC = total cholesterol, TG = triglycerides, TNF-α = tumor necrosis factor alpha, VF = visceral fat, Wt = weight.

**Table 1 nutrients-16-02941-t001:** Primary constituents in Chios mastic gum.

Fractions	Compound Names
**Essential oil**	Tricyclene, α-pinene, camphene, verbenene, β-pinene, β-myrcene, 2-methylanisole, p-cymeme, limonene, *trans*-linool oxide, α-campholene aldehyde, *trans*-pinocarveol, *trans*-verbenol, pinocamphone, pinocarvone, p-mentha-1,5-dien-8-ol, myrtenal, myrtenol, verbenone, β-carophyllene, α-caryophyllene, carophyllene oxide
**Polymer**	cis-1,4-poly-β-myrcene
**Triterpene**	Oleanonic acid, oleanolic acid, oleanonic aldehyde, oleanolic aldehyde, moronic acid, 28-nor-oleanone, 28-nor-aleanole, 28-hydroxy-β-amyrone, β-amyrine, β-amyrone, germanicol, lupeol, betulonal, lup-20(29)-ene-3-one, 3-oxo-28-norlup-20(29)-ene, 24*Z*-masticadienonic acid, 24*Z*-isomasticadienonic acid, 24*Z*-masticadienolic acid, 24*Z*-isomasticadienolic acid, mastichadienonal, isomastichadienolal, tirucallol, dammaradienone, mastichinonic acid, butyrospermol, 20(*S*)-3β-acetoxy-20-hydroxydammar-24-ene, dipterocarpol, 3β-hydroxymalabarica-14(26),17*E*,21-triene, 3-oxomalabarica-14(26),17*E*,21-triene, (8*R*)-3β,8-dihydroxy-polypoda-13*E*,177E,21-triene, (8*R*)-3-oxo-8-hydroxypolypoda-13*E*,17*E*,21-treiene

**Table 2 nutrients-16-02941-t002:** Human interventions with Chios mastic gum and outcomes related to cardiometabolic risk factors.

Reference	Population	Duration and Design	Outcomes
**Gioxari [53]**	Participants with metabolic disorders (*n* = 94)	3 months: CMG EO (adjunct metabolic disease treatment) vs. control	Treatment group: ↓ TG, LDL, oxLDL, SBP, ALT, AST, GGT, weight, BMI, % body fat and visceral fat, and ↑ adiponectin
**Fukazawa [51]**	Healthy male Japanese (*n* = 21)	6 months: 3 groups (control, CMG, and CMG + PA)	At 3 months, CMG and CMG + PA: ↓ TG; at 3 and 6 months, CMG + PA ↓ insulin and HOMA-IR; at 6 months CMG ↓ insulin and HOMA-IR
**Triantafyllou [50]**	Healthy volunteers (*n* = 133)	18 months high dose vs. 12 months low dose	High dose: ↓ TC, LDL, TC/LDL ratio, lipoprotein (a) apolipoprotein B
**Kanoni [57]**	Obese individuals with NAFLD (*n* = 98)	6 months: CMG or placebo	Patients with BMI > 35 kg/m ^2^ had higher total antioxidant status; CMG–gene interactions were observed with cytokines and antioxidant biomarkers
**Amerikanou [56]**	Patients with NAFLD and obesity (*n* = 98)	6 months: CMG or placebo	↓ inflammatory markers and liver inflammation fibrosis score only in patients with BMI > 35 kg/m^2^; improved microbiota and lipid metabolite profiles
**Kartalis [52]**	Individuals with TC > 200 mg/dL (*n* = 156)	8 weeks: 4 groups (placebo, total mastic, polymer-free mastic, and powder mastic)	In total mastic group only: ↓ TC, fasting blood glucose (the effect was stronger with BMI > 25 kg/m^2^)
**Kontogiannis [61]**	*n* = 27 individuals, *n* = 13 with high blood pressure	Allocated to CMG or placebo. Analysis taken one week apart.	In hypertensive patients, ↓ a/pSBP, pPP; downregulation of the proteostatic and NOx2 pro-oxidant pathway

Abbreviations used: ALT = alanine aminotransferase, AST = aspartate aminotransferase, CMG = Chios mastic gum, EO = essential oil; GGT = gamma-glutaryl transferase, HOMA-IR = homeostatic model assessment for insulin resistance, LDL = low density lipoprotein, oxLDL = oxidized LDL; NOx2 = NADPH oxidase, PA = physical activity, pPP = peripheral pulse pressure, SBP = systolic blood pressure (a/p refers to aortic or peripheral), TC = total cholesterol, TG = triglycerides.

**Table 3 nutrients-16-02941-t003:** Animal studies with Chios mastic gum and outcomes related to cardiometabolic risk factors.

Reference	Animal Model	Duration and Design	Outcomes
**Georgiadis [63]**	Streptozotocin induced diabetic C57BL/6 mice and controls (*n* = 27)	8 weeks: 3 groups (control, low-dose mastic, high-dose mastic)	Low-dose mastic group: ↓ glucose, TC, LDL, TG, and ↑ HDL; high dose exhibited ↓ TG; partially reversed hepatic steatosis in both groups
**Tzani [66]**	2K1C hypertensive rats and sham (*n* = 25)	2 weeks: CMG treatment or control	When comparing 2K1C to 2K1C + CMG rats after CMG administration: ↓ SBP, DBP, mean arterial pressure, renin, CRP, IL-6; improved heart biomechanical indices
**Vallianou [62]**	Hyperlipidemic-induced and naïve rats (*n* = 18)	24 h: 2.5%, 4%, 5%, or 7.5% CMG EO and CMG EO constituents	CMG EO administration in naïve rats resulted in ↓ TC, LDL, TG; CMG EO treatment in hyperlipidemic rats resulted in ↓ TC, LDL, TG; camphene in hyperlipidemic rats resulted in ↓ cholesterol, LDL, TG
**Kannt [65]**	NASH-induced C57BL/6J mice and controls (*n* = 34)	8 weeks: 3 groups (lean chow, DIO-NASH, DIO-NASH + CMG)	CMG supplementation led to ↓ ALT, liver TC, total lipids, fibrosis, histological NAFLD activity score; ↑ in gut microbiota diversity
**Andreadou [64]**	Rabbits (*n* = 43)	6 weeks: 6 groups (SFO controls, ME, and MN across normal-fed and cholesterol-enriched diets)	ME and MN both ↓ infarct size and ↓ MDA in normal-fed rabbits only, hypercholesterolemic rabbits had ↓ TC and LDL when treated with ME and MN

Abbreviations used: 2K1C = 2-kidney, 1-clip, CMG = Chios mastic gum; DBP = diastolic blood pressure, DIO = diet-induced obesity; EO = essential oil; HDL = high density lipoprotein, MDA = malondialdehyde, ME = mastic extract without polymer, MN = neutral mastic fraction, NASH = non-alcoholic steatohepatitis, SFO = sunflower oil; TC = total cholesterol; TG = triglycerides

**Table 4 nutrients-16-02941-t004:** Proposed mechanisms of action and outcomes from in vitro and in silico studies with Chios mastic gum.

Reference	Pathway/Mechanism Influenced	Outcomes
**Loizou [67]**	Adhesion molecules, monocytes, NF-κB	↓ adhesion molecule expression, ↓ monocyte attachment, ↓ phosphorylation of NF-κB p65
**Vallianou [62]**	HMG-CoA ruled out	↓ cholesterol, no HMG-CoA reductase activity
**Xanthis [68]**	Cell migration, cytoprotection, antioxidant, NRF-2	↑ mRNA of antioxidant genes, ↑ cell viability on oxidative stressor exposure, ↑ wound closure
**Vuorinen [71]**	GR	CMG triterpenoids inhibit 11β-HSD1
**Peterson [69]**	PPARγ	Oleanolic acid and other CMG subfractions exhibit PPARγ agonism
**Dedoussis [72]**	Antioxidant (GSH), oxLDL (CD36)	Triterpenoids ↑ intracellular glutathione, ↓ CD36 mRNA, ↓ oxLDL
**Andrikopoulos [73]**	oxLDL	↓ oxLDL
**Kalousi [74]**	GR, AMPKα, PEPCK, PPARα, NF-κΒ, apoptosis	↓ cell viability and ↑ apoptosis, ↓ transcriptional activation and proteins of multiple metabolic sensors, ↓ inflammation

Abbreviations used: 11β-HSD1 = 11β-hydroxysteroid dehydrogenase, AMPK = adenosine monophosphate-activated protein kinase, CD36 = cluster of differentiation 36, GR = glucocorticoid receptor, NF-κB = nuclear factor kappa-light-chain-enhancer of activated B cells, NRF-2 = nuclear factor erythroid 2-related factor 2, PPARγ/α = peroxisome proliferator-activated receptor gamma/alpha, PEPCK = phosphoenolpyruvate carboxykinase, oxLDL = oxidized low-density lipoprotein.

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
