# Peer review of "Chios Mastic Gum: A Promising Phytotherapeutic for Cardiometabolic Health"

_nutrients, 2024, doi:10.3390/nu16172941_

Round 1

Reviewer 1 Report

Comments and Suggestions for Authors

The narrative review "Chios Mastic Gum: A Promising Phytotherapeutic for Cardiometabolic Health" comprehensively examines Chios mastic gum (CMG), a resin extracted from the Pistacia lentiscus var. Chia tree, which is known for its long history of medicinal use. The review's primary focus is to explore CMG's bioactive components and their potential therapeutic roles in managing cardiometabolic diseases (CMD). With CMD being a major global health concern, this review is timely and relevant, aiming to consolidate the evidence surrounding CMG's beneficial properties, particularly its anti-inflammatory, antioxidant, and metabolic effects.

The authors said that they included " 20 articles for the narrative literature review" but the references are more than 20. What are the limitations?

One of the main weaknesses highlighted in the review is the relatively limited number of studies, particularly high-quality clinical trials, exploring CMG’s effects on CMD. The mixed results across some studies underscore the need for more rigorous research to confirm the therapeutic efficacy of CMG. Additionally, the review could benefit from a more in-depth discussion of the limitations of existing studies, such as small sample sizes, short duration, and potential confounding factors.

Further exploration of the synergistic effects of CMG’s bioactive compounds and their interactions with other phytochemicals or pharmaceuticals is also suggested. Additionally, expanding research on the effects of CMG on gut microbiota and its potential to influence CMD through the gut-heart axis represents a promising avenue for future studies.

The review is supported by three tables that summarize key data and one figure, but it does not include any other figures, which might limit the visual representation of complex interactions and findings.

A table that summarizes the various bioactive compounds found in CMG, including terpenes, polyphenols, and flavonoids must be added. This table can serve as a quick reference for understanding the primary constituents responsible for CMG's health benefits.

The potential mechanisms through which CMG may exert its therapeutic effects on CMD were not discussed. The mechanisms under headings such as lipid metabolism, glucose regulation, inflammation, oxidative stress, and microbiota modulation can be considered.

Author Response

Thank you for taking the time to review this manuscript. Please find the detailed responses below and the corresponding revisions/corrections highlighted/in track changes in the re-submitted files.

Comments 1: The authors said that they included " 20 articles for the narrative literature review" but the references are more than 20. What are the limitations?

Response 1:

We are not sure what the reviewer means by this. We included the most recent references related to this narrative. These are the ones that we found in English, that were recent and related to cardiometabolic problems. We did not include related to other diseases, and this may be the reason why the reviewer is talking about limitations.

Comments 2: One of the main weaknesses highlighted in the review is the relatively limited number of studies, particularly high-quality clinical trials, exploring CMG’s effects on CMD. The mixed results across some studies underscore the need for more rigorous research to confirm the therapeutic efficacy of CMG. Additionally, the review could benefit from a more in-depth discussion of the limitations of existing studies, such as small sample sizes, short duration, and potential confounding factors.

Response 2: Thanks for your reflection on the weaknesses of the current studies, and for some specific suggestions about what else should be added to the discussion of limitations. As per your suggestion, we have expanded the limitations section and included the specific language you recommended. You can find these changes starting at like 340 and 352.

“Shorter study duration and smaller sample sizes are among the challenges in interpreting these studies.”

“….dietary intake and physical activity should also be accounted for in future clinical trials, as they are significant confounders. A possible method to potentiate the positive impacts of CMG, however, may include utilizing it alongside Mediterranean-style, plant-based, or Nordic dietary patterns.

Other lifestyle factors (such as smoking, alcohol consumption, sleep patterns), the dosing regimen of CMG, interactions with potential medications, and underlying health conditions are additional confounders that have yet to be fully acknowledged and addressed in these clinical trials. These mixed results and study designs underscore the need for more rigorous research to validate the therapeutic efficacy of CMG.”

Comments 3: Further exploration of the synergistic effects of CMG’s bioactive compounds and their interactions with other phytochemicals or pharmaceuticals is also suggested. Additionally, expanding research on the effects of CMG on gut microbiota and its potential to influence CMD through the gut-heart axis represents a promising avenue for future studies.

Response 3: Agreed. We appreciated some of your language here so included it in the manuscript as well, which can be found starting on lines 389 and 437.

“CMG’s potential interactions with other phytochemicals and pharmaceuticals may be another fruitful route of investigation.”

“Elucidating the role of the microbiota may ultimately provide valuable insights into the impacts of the gut-heart axis.”

Comments 4: The review is supported by three tables that summarize key data and one figure, but it does not include any other figures, which might limit the visual representation of complex interactions and findings.

Response 4:  We do not believe that another figure would better highlight this review. We have included a great deal of information in the Tables and our narrative explains the interactions and findings.

\

Comments 5: A table that summarizes the various bioactive compounds found in CMG, including terpenes, polyphenols, and flavonoids must be added. This table can serve as a quick reference for understanding the primary constituents responsible for CMG's health benefits.

Response 5: Great suggestion, thank you. You can find this table of primary constituents at the bottom of page 2, starting on line 87.

Comments 6: The potential mechanisms through which CMG may exert its therapeutic effects on CMD were not discussed. The mechanisms under headings such as lipid metabolism, glucose regulation, inflammation, oxidative stress, and microbiota modulation can be considered.

Response 6: Thanks for pointing this out. You can find an expanded discussion on mechanisms, as elucidated by Tables 2-4, starting at line 392.

For metabolic and microbiota modulation, human studies have revealed CMG reduces TG, LDL, TC, insulin, fasting glucose, HOMA-IR, Lp(a), Apo(B), LPC, LPE, and cholic acid. Increases in HDL and microbiota diversity were also observed. Adjunct metabolic syndrome treatments, there was a reduction in body fat, visceral fat, and weight, and an increase in adiponectin. CMG-gene interactions also revealed reductions in hemoglobin. Animal studies validate some of these findings, with reductions in glucose, TG, LDL, TC, and total lipids, and increases in HDL and microbiota diversity. In vitro and in silico studies reveal these results may be partially mediated through the reduction in activity of various pathways and receptors such as PEPCK, GR, PPARγ/α, AMPKα, and 11β-HSD1.

For cardiovascular and hepatic impacts in humans, studies have shown CMG supplementation reduces SBP and pPP, while adjunct metabolic syndrome treatment studies have also shown additional reductions in ALT, AST, and GGT. Animal studies fortify these findings with the observed reductions in hepatic steatosis, NAFLD score, liver fibrosis, ALT, SBP, DBP, MAP, renin, and infarct size. Improvements in numerous cardiac indices have also been observed with CMG supplementation. Modulation in oxidative stress and antioxidant genes and pathways has been observed in humans because of CMG supplementation, such as reductions in oxLDL and NOS-2. CMG-gene interaction studies have revealed a differential impact on glutathione peroxidase (Gpx) levels (increase or decrease), and an increase in total antioxidant status (TAS). In vitro studies support these studies with reductions in oxLDL, NRF-2, CD36 and GSH.

Finally, human studies have also shown that CMG supplementation impacts inflammation and immunity particularly through the CMG-gene axis, where a reduction in IL-6 and TNF-α and their genes were observed, as well as a reduction in IL-10. In animal studies, decreases in CRP and IL-6 have been observed, while in vitro studies have demonstrated decreases in monocyte attachment, adhesion molecules, NF-κΒ, p65, and cell migration. Notably, modulation of oxidative stress, antioxidants, inflammation and immunity presumably is enough to drive the cardiovascular, hepatic, and metabolic outcomes observed in human studies.”

Reviewer 2 Report

Comments and Suggestions for Authors

In the present narrative review, Sarah A. Blomquist and Maria Luz Fernandez briefly overviewed the bio- active constituents of chios mastic gum (CMG), a resin obtained from the Pistacia lentiscus var. Chia tree, and examined and described its potential as a cardiometabolic disease (CMD) phytotherapeutic. Specifically, the authors concluded that, despite relatively limited studies with mixed results, CMG may be a unique phytotherapeutic for improving biomarkers related to lipid and glucose metabolism, cardiovascular and hepatic health, inflammation, oxidative stress, body composition, and microbiota. Overall, I think that the manuscript is well-structured (within the scope of "Nutrients”) and of clinical impact on a current topic of interest. I have some small suggestion/curiosity to improve the quality of review.

1.       Please carefully check if it has been reported any side effect after CMG supplementation. It could be useful add a specific comment about this point.

2.       GLUT-1 and GLUT-4 seem to be involved in the pathophysiological mechanism of cardiometabolic diseases. This feature could be carefully evaluated and eventually discussed.

3.       The authors should keep in mind that the dietary pattern of the patients (i.e. Mediterranean-style diet, Plants-based diet, Nordic dietary pattern, etc.) can strongly influence the overall results related to CMG supplementation in the alleviation of CMD.

4.       The authors could add in Graphical form the association between dietary patterns, food components and CMG; in this way, I feel that the readers can better understand its possible therapeutic potential as a cardiometabolic disease (CMD) phytotherapeutic.

Thank you for asking me to review this manuscript. Overall, I think that the paper is well-structured, and it could be of interest to the readers of "Nutrients”. It deserves to be published after minor revisions.

Comments on the Quality of English Language

Minor editing of English language is required.

Author Response

1. Summary

Thank you for taking the time to review this manuscript. Please find the detailed responses below and the corresponding revisions/corrections highlighted/in track changes in the re-submitted files.

Comments 1:  Please carefully check if it has been reported any side effect after CMG supplementation. It could be useful add a specific comment about this point.

Response 1: We agree with this comment. We have added this information to the review paper, which can be found starting at line 182.

“Notably, none of these clinical trials reported any side effects with CMG supplementation, making it a safe and potentially desirable therapeutic.”

Comments 2: GLUT-1 and GLUT-4 seem to be involved in the pathophysiological mechanism of cardiometabolic diseases. This feature could be carefully evaluated and eventually discussed.

Response 2: We appreciate this suggestion and agree, however none of the studies we found examined GLUT-1 and GLUT-4 as potential mechanisms through which CMG exerts its action. Therefore, we feel it is outside the scope of this review.

Comments 3: The authors should keep in mind that the dietary pattern of the patients (i.e. Mediterranean-style diet, Plants-based diet, Nordic dietary pattern, etc.) can strongly influence the overall results related to CMG supplementation in the alleviation of CMD.

Response 3: Thank you for this suggestion. We have added comments to the manuscript addressing how beneficial dietary patterns may potentiate the impacts of CMG on CMD outcomes. This can be found starting at line 353.

“A possible method to potentiate the positive impacts of CMG, however, may include utilizing it alongside Mediterranean-style, plant-based, or Nordic dietary patterns.”

1. Summary

Thank you for taking the time to review this manuscript. Please find the detailed responses below and the corresponding revisions/corrections highlighted/in track changes in the re-submitted files.

Comments 1:  Please carefully check if it has been reported any side effect after CMG supplementation. It could be useful add a specific comment about this point.

Response 1: We agree with this comment. We have added this information to the review paper, which can be found starting at line 182.

“Notably, none of these clinical trials reported any side effects with CMG supplementation, making it a safe and potentially desirable therapeutic.”

Comments 2: GLUT-1 and GLUT-4 seem to be involved in the pathophysiological mechanism of cardiometabolic diseases. This feature could be carefully evaluated and eventually discussed.

Response 2: We appreciate this suggestion and agree, however none of the studies we found examined GLUT-1 and GLUT-4 as potential mechanisms through which CMG exerts its action. Therefore, we feel it is outside the scope of this review.

Comments 3: The authors should keep in mind that the dietary pattern of the patients (i.e. Mediterranean-style diet, Plants-based diet, Nordic dietary pattern, etc.) can strongly influence the overall results related to CMG supplementation in the alleviation of CMD.

Response 3: Thank you for this suggestion. We have added comments to the manuscript addressing how beneficial dietary patterns may potentiate the impacts of CMG on CMD outcomes. This can be found starting at line 353.

“A possible method to potentiate the positive impacts of CMG, however, may include utilizing it alongside Mediterranean-style, plant-based, or Nordic dietary patterns.”

Comments 4: The authors could add in Graphical form the association between dietary patterns, food components and CMG; in this way, I feel that the readers can better understand its possible therapeutic potential as a cardiometabolic disease (CMD) phytotherapeutic.

Response 4: We thank you for the suggestion but feel, however, that focusing on dietary patterns is outside the scope of our paper.

Round 2

Reviewer 1 Report

Comments and Suggestions for Authors

I agree.